# Assessment of Particulate Matter, Heavy Metals, and Carbon Deposition Capacities of Urban Tree Species in Tehran, Iran

Sahar Elkaee [1,2], Anoushirvan Shirvany [1,*], Mazaher Moeinaddini [3] and Farzaneh Sabbagh [4,*]

1 Department of Forestry and Forest Economics, Faculty of Natural Resources, University of Tehran, Karaj 31587-77871, Iran; elkaee.sa@o.cnu.ac.kr
2 Department of Environmental & IT Engineering, Chungnam National University, 99 Daehak-ro, Yuseong-gu, Daejeon 34134, Republic of Korea
3 Department of Environmental Science, Faculty of Natural Resources, University of Tehran, Tehran 14179-35840, Iran; moeinaddini@ut.ac.ir
4 Department of Plant Sciences, Faculty of Biological Science, Alzahra University, Tehran 19938-91176, Iran
* Correspondence: shirvany@ut.ac.ir (A.S.); farzaneh2464@gmail.com (F.S.)

**Abstract:** Air pollution is a pressing environmental concern in urban areas, with particulate matter (PM) posing serious health and environmental threats. Urban greening has emerged as a potential solution to capture and retain PM. This study assesses the PM deposition capacity of five common tree species: *Morus alba* (*M. alba*), *Ailanthus altissima* (*A. altissima*), *Platanus orientalis* (*P. orientalis*), *Robinia pseudoacacia* (*R. pseudoacacia*), and *Ulmus minor* (*U. minor*) in two highly polluted sites in Tehran, Iran. Additionally, this study investigates the accumulation of heavy metals (Ni, Fe, Cd, and Pb), Organic Carbon (OC), Elemental Carbon (EC), and Total Carbon (TC) on the leaves of these tree species. The results demonstrate species-specific differences in PM deposition capacity, with *U. minor* and *M. alba* showing high PM retention. *A. altissima* exhibits strong capability in adsorbing PM 0.1–2.5, while *U. minor* demonstrates greater retention of PM > 2.5. Moreover, the deposition of heavy metals varies among species, with *R. pseudoacacia* and *A. altissima* capturing higher levels. This study highlights the significance of appropriate tree utilization in urban environments against air pollution in order to make the air healthier in major cities. Awareness of the different tree species capacities leads urban planners and policymakers to make intelligent decisions about urban greening initiatives to improve air quality and overall well-being.

**Keywords:** air pollution; particulate matter; urban greening; tree species; air quality management

## 1. Introduction

Air pollution has emerged as a pressing global environmental issue, particularly in urban areas where high population densities and extensive anthropogenic activities contribute to elevated levels of pollutants in the atmosphere. The presence of particulate matter (PM) stands out among the pollutants, encompassing a diverse range of suspended solid or liquid particles with varying sizes and chemical compositions [1–3]. This mixture includes heavy metals, black carbon, polycyclic aromatic hydrocarbons, and other predominantly human-made pollutants suspended in the atmosphere [4,5].

Particles are categorized into three groups based on their sizes: coarse (2.5–10 µm), fine (0.1–2.5 µm), and ultrafine (≤0.1 µm) [3,6]. The repercussions of these minuscule particles reverberate through both human health and the environment. They have a significant role in the development of respiratory and cardiovascular diseases in humans while also inflicting harm upon ecosystems and climate [2,7–9]. PM aerosol, originating from various sources like coal, biomass, and industrial activities, possesses a complex composition, including carbonaceous species, soot, inorganic ions, and elements. Among these, polycyclic aromatic hydrocarbons (PAHs) and certain elements (Cd, Pb, Ni, As, Tl, and Hg) pose health risks due to their carcinogenic and mutagenic properties. PAHs form during incomplete

combustion, while elements are emitted from sources like coal combustion and traffic. The potential harm arises when PAHs and elements in PM deposit on the lungs, dissolving into the pulmonary surfactant [10–12]. It is crucial to delve into the specific sources of carbon, PAHs, and heavy metals in developing countries, where PM concentrations are notably high, emphasizing their potential impact on human health. Additionally, recognizing the significance of greenery in urban areas for capturing PM is essential for a holistic understanding of the dynamics involved.

Particulate carbon, such as Organic Carbon (OC), Elemental Carbon (EC), and Total Carbon (TC), mainly in the form of Carbonate Carbon (CC), is formed through the photo-oxidation and polymerization of organic species in the environment [13–15]. EC, in particular, results from the incomplete combustion of carbon-based fuels, found in wood used for heating and fossil fuels used in various industrial and transportation activities [13,14]. Health concerns surrounding automobile pollution, particularly in the context of elemental carbon dominating total carbon emissions from specific vehicles, have been raised [16–18].

Heavy metals, some of which occur naturally in soils, rocks, waters, and organisms, and others stemming from human activities, have been widely studied for their impact on plants and the environment [19,20]. While certain heavy metals pose substantial environmental hazards, others are essential nutrients for living organisms but can become toxic at elevated levels or in specific forms [21–23]. The accumulation of heavy metals in biological organisms and soil poses significant risks, particularly lead, which tends to accumulate on urban roadways due to traffic emissions. Industrial and automobile emissions further impact the composition of soil, water, and air [24–27].

Recent attention has focused on the concept of "urban greening", which involves using vegetation as a natural solution to mitigate air pollution [8,9,28–30]. Trees and plants can serve as natural filters, capturing and retaining particulate matter on their surfaces [31,32]. This unique capability of vegetation to act as "green lungs" for urban environments has spurred interest in exploring the potential of different tree species to combat air pollution and improve the well-being of urban inhabitants [6,33,34].

The influence of the surface properties of objects on particle immobilization has been well-established, and different plant species exhibit varying capabilities in scavenging dust-laden air [35]. The dust-retention capacities of vegetation are contingent on several factors, encompassing canopy type, leaf and branch density, and leaf micromorphology, such as roughness, trichomes, and wax [33,36–38]. Variations in leaf surface properties within species cultivars can also significantly impact their ability to capture PM. To comprehensively understand the relationships between species traits and their particulate matter-capturing capacity, large-scale sampling efforts are imperative [33]. While it is recognized that urban trees temporarily reduce atmospheric PM concentrations by retaining particles, the efficacy of vegetation as a long-term alternative to other mitigation measures remains a topic of ongoing debate [33,39].

Given Tehran's ongoing air pollution challenges due to high population density, rapid urbanization, vehicular emissions, and industrial activities, the city serves as an ideal case study for urban air quality management [40–42]. Additionally, the presence of the Alborz Mountain range surrounding Tehran restricts air circulation, leading to the accumulation of pollutants in certain areas. These complex geographical and climatic conditions offer an opportunity to evaluate the performance of various tree species in capturing particulate matter and heavy metals, which are major contributors to air pollution [43–45].

This study aims to assess the PM deposition capacity of five common tree species—*Morus alba* (*M. alba*), *Ailanthus altissima* (*A. altissima*), *Platanus orientalis* (*P. orientalis*), *Robinia pseudoacacia* (*R. pseudoacacia*), and *Ulmus minor* (*U. minor*)—in two highly polluted sites within Tehran. By analyzing the potential of these tree species to capture particulate matter in three size fractions (<0.1 μm, 0.1–2.5 μm, and >2.5 μm), this study seeks insights into their effectiveness as natural air cleaner filters. Additionally, the deposition of heavy metals (Ni, Fe, Cd, and Pb) and EC, OC, and TC on the leaves of these species will be investigated,

considering that heavy metals and carbon are often associated with particulate matter and can have serious health and environmental implications.

## 2. Materials and Methods

### 2.1. Overview of the Study Area

This study was conducted in Tehran, the capital city of Iran, positioned at approximately latitude 35°50′ N and longitude 51°37′ E and situated at an average elevation of around 1400 m above sea level (Figure 1). Tehran experiences a broad annual temperature range, with January recording an average low temperature of 4 °C and August reaching an average high temperature of 37 °C. Daytime relative humidity in Tehran varies significantly, from a minimum of 28% in June to a maximum of 64% in December. The presence of the Alborz Mountain range to the north and northeast of the city restricts the dispersion and ventilation of air pollutants. Tehran, ranking 19th globally in terms of population, confronts substantial air pollution challenges, primarily stemming from sources such as vehicular emissions [46].

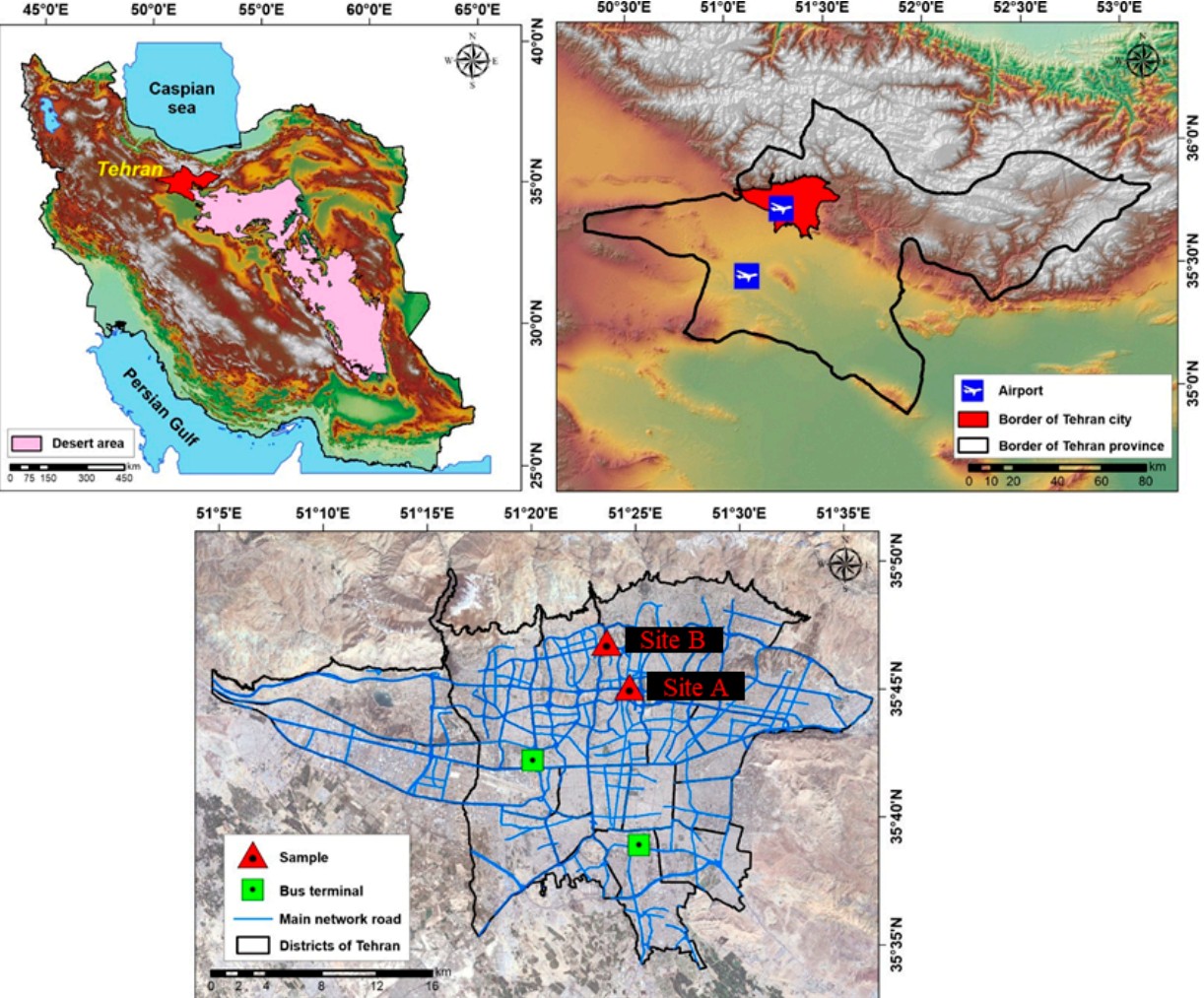

**Figure 1.** A map illustrating the geographical locations of the study areas in Tehran, Iran. The map specifically highlights Chamran Highway, denoted as 'site B', and Gandi Street, identified as 'site A'. Site A is situated along a residential-commercial road with comparatively lower traffic densities, while Site B is located within a densely urbanized area characterized by high traffic volumes.

## 2.2. Sampling and Analytical Methods

Sampling locations were meticulously chosen based on road types in proximity to each Air Quality Monitoring Station (AQMS), following an extensive analysis of long-term PM concentration data (Figure 1). Two highly polluted sites, identified during on-site visits, were deemed suitable for collecting samples of PM deposition on tree leaves. These sites, situated along Chamran Highway (referred to as 'site B') and Gandi Street (referred to as 'site A'), are characterized as residential-commercial thoroughfares.

A total of 30 samples were gathered from the most prevalent tree species found at these locations, including *M. alba*, *A. altissima*, *P. orientalis*, *R. pseudoacacia*, and *U. minor*. Random sampling was conducted, ensuring a minimum distance from the street. Three individuals from each tree species were chosen, and leaves were randomly collected from four directions (east, south, west, and north) at a height of 2.0–2.5 m above the ground. Fifteen leaf pieces were collected in each direction for every species, resulting in sixty pieces per species (180 pieces in total for three plants) [47]. The sampling took place in the latter part of September, following the conclusion of the growing season, with a gap of over two months since the last rainfall event. There were no severe weather conditions, such as heavy rains or strong winds, in the week leading up to the collection. On the sampling day, the weather was sunny, and the wind speed remained below 2 m/s to prevent errors caused by external factors [5,46].

To prevent contamination, the collected leaves were meticulously labeled and securely sealed in zip-lock bags during the sampling process, ensuring no particle loss. These samples were then preserved in a laboratory refrigerator at a low temperature (4 °C) [48]. Subsequently, the leaves were cleansed with ethanol and distilled water to achieve samples in a 1.5:10 mL ratio. The ethanol in the eluted samples was evaporated in an oven set at 70 °C. The quantity of deposited PM was determined through centrifugation (utilizing the German-manufactured 2206A model) for 5 min at 3000 rpm, and the upper liquid layer was removed. The samples were subsequently oven-dried at 105 °C [46]. The suspended particles were allowed to settle, and water was removed using a pipette. The resultant particles were then placed in an oven at 135 °C for a duration of 24 h. Finally, the dry particles were weighed using a digital scale with a precision of 0.001 g. The leaves of each species were dried and scanned, and both the leaf area and average leaf area for each species were estimated using Image J software (Image J 1.44p) (Figure 2) [49].

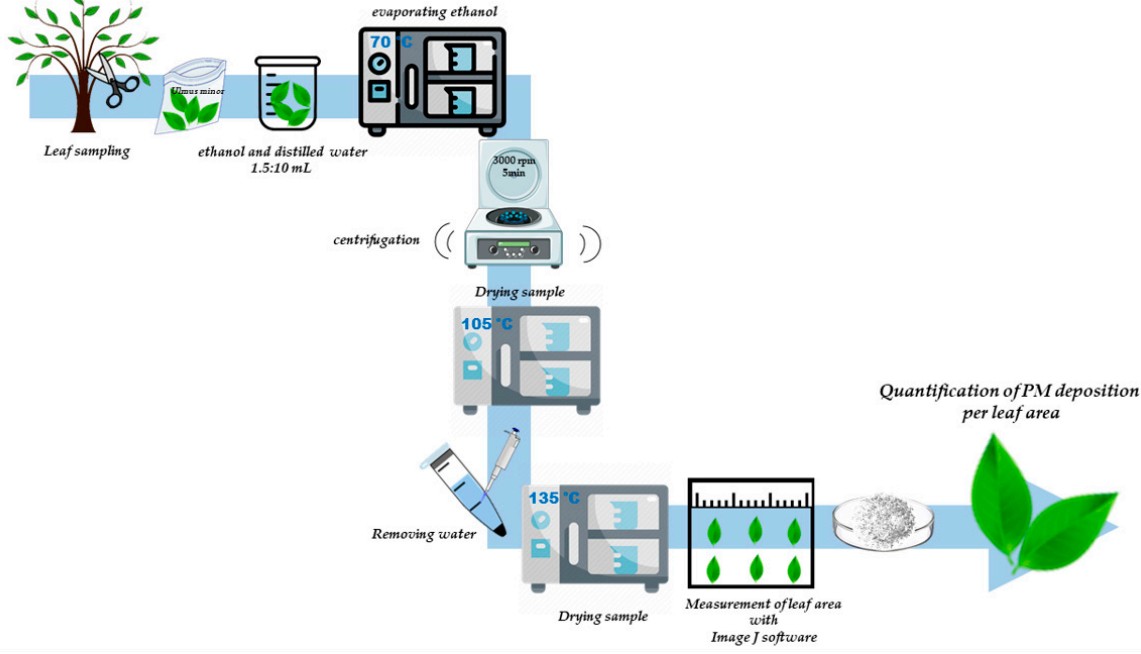

**Figure 2.** Schematic representation of particulate matter (PM) deposition quantification per leaf area.

For the determination of Total Carbon (TC) in the dry samples, a controlled process was employed. The samples underwent combustion at 550 °C within a muffle furnace for a duration of 2 h. This specific temperature was chosen because carbonates remain stable at temperatures below 550 °C [50]. Subsequent to the 2 h combustion period, the samples were transferred to an oven maintained at 105 °C for several hours to ensure complete drying. Once the samples had cooled within desiccators, their weight was recorded. The calculation of loss-on-ignition was performed using Formula (1) [50]:

$$\frac{\text{dry samples mass before combustion} - \text{dry sample mass after combustion}}{\text{dry sample mass before combustion}} \times 100 \quad (1)$$

The Walkley and Black method was employed to determine the Organic Carbon (OC) content. In this method, a measured sample portion (0.3 g) was treated with 10 CC of a 0.4 N potassium dichromate solution ($K_2Cr_2O_7$), followed by the addition of 20 CC of concentrated sulfuric acid. Subsequently, 100 CC of triple-distilled water were gently stirred into the mixture, which was then placed in a fume hood at room temperature for 16 to 18 h. To quantify the excess dichromate, a standard 0.2 N ferrous ammonium sulfate ($Fe(NH_4)_2(SO_4)_2 \cdot 6H_2O$) solution was used for potentiometric back-titration [51].

The Elemental Carbon (EC) content was determined by subtracting the Organic Carbon from the Total Carbon, following Formula (2) [52]:

$$TC = OC + EC \quad (2)$$

For the extraction of heavy metals (Fe, Ni, Cd, and Pb) from the dry samples, 0.5 g of PM was introduced into a 25-mL laboratory flask, and the wet digestion method (utilizing $HNO_3$–HCl in a 4:1 ratio) was employed. Heavy metal analysis was conducted using an atomic absorption/flame spectrophotometer (Shimadzu, AA-670, Kyoto, Japan) [46].

The particle dimensions of the PM collected from the leaf surfaces of these species were determined utilizing a particle size analyzer device (Brookhaven MAS-BI/90Plus Particle Size Analyzer, New York, NY, USA).

SPSS version 17 statistical software were utilized for data analysis. Two-sample independent *t*-tests, Games-Howell tests, and Duncan tests were performed to assess the distinctions between the samples. Any difference with a *p*-value of less than 0.05 was considered statistically significant.

## 3. Results

*3.1. Particulate Matter (PM) Deposition Capacity of Different Tree Species in Two Distinct Sites*

This study delves into the assessment of various tree species' proficiency in capturing particulate matter (PM) in two distinct locations, denoted as Site A (Gandi Street) and Site B (Chamran Highway). Site A is situated along a residential-commercial road with comparatively lower traffic densities, while Site B is located within a densely urbanized area characterized by high traffic volumes.

The data depicting PM deposition on the leaves of five tree species at both Site A and Site B are presented in Figure 3. It is evident that at Site A, *U. minor* demonstrated notably higher PM deposition, followed by *M. alba*, *P. orientalis*, and *R. pseudoacacia*, with *A. altissima* exhibiting the lowest PM deposition. At Site B, the order of PM deposition was as follows: *M. alba* > *U. minor* = *P. orientalis* > *A. altissima* = *R. pseudoacacia*. These findings highlight the strong capability of *U. minor* in Site A and *M. alba* in Site B to effectively capture PM on their leaves (Figure 3).

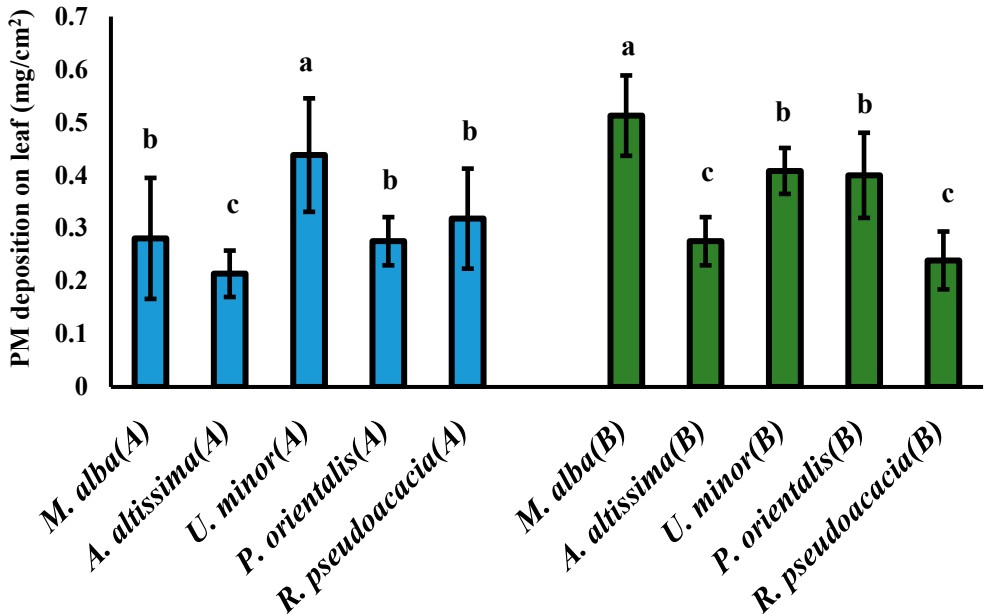

**Figure 3.** Comparison of total PM deposition on leaves among five species in Sites A and B: mean ± standard deviation with significance ($p < 0.05$). Different lowercase letters indicate significant differences.

A comparison of PM deposition capacities at Site A and Site B revealed significant differences in the PM-capturing abilities of *M. alba* and *P. orientalis*. *M. alba* showed a higher PM deposition rate in Site B ($0.512 \pm 0.075$ mg/cm$^2$) than in Site A ($0.280 \pm 0.114$ mg/cm$^2$), while *P. orientalis* exhibited a greater PM deposition rate in Site B ($0.399 \pm 0.080$ mg/cm$^2$) than in Site A ($0.275 \pm 0.045$ mg/cm$^2$).

*3.2. Particulate Matter Distribution on Leaves of Tree Species across Two Sites in Three Particle Size Fractions (<0.1 µm, 0.1–2.5 µm, >2.5 µm)*

The examination of particulate matter (PM) deposition on leaves within various particle size fractions (<0.1 µm, 0.1–2.5 µm, and >2.5 µm) unveiled notable variations among tree species at Sites A and B (Table 1).

**Table 1.** Particle size fractions of PM deposition by various species in Sites A and B presented as percentage frequencies.

| Species | Frequency % | | | |
|---|---|---|---|---|
| | Site A | Site B | Site A | Site B |
| | 0.1–2.5 µm | | >2.5 µm | |
| *R. pseudoacacia* | 99.40% | 65.00% | 0.60% | 35.00% |
| *U. minor* | 100% | 32.70% | 0% | 67.30% |
| *M. alba* | 66.70% | 44.20% | 33.30% | 55.80% |
| *A. altissima* | 99.40% | 100% | 0.60% | 0% |
| *P. orientalis* | 100% | 59.70% | 0% | 40.30% |

Table 1 provides a comprehensive comparison of PM deposition in three distinct particle size fractions at Sites A and B for various tree species. The results underscore the prevalence of PM in the 0.1–2.5 µm fraction at Site A, while Site B exhibits a higher presence of PM exceeding 2.5 µm.

In both Site A and Site B, none of the species exhibited a presence of particulate matter below 0.1 µm (PM < 0.1). At Site A, *U. minor* and *P. orientalis* displayed a remarkable 100% reduction in PM within the 0.1–2.5 µm fraction compared to *R. pseudoacacia*, *A. altissima*

(99.4%), and *M. alba* (66.7%). Furthermore, *M. alba* exhibited a 33.3% reduction in PM exceeding 2.5 μm, while *R. pseudoacacia* and *A. altissima* showed a minimal reduction of 0.6% (Table 1).

Transitioning to Site B, the concentrations of PM exceeding 2.5 μm were notably higher in *P. orientalis*, *M. alba*, and *U. minor*, accounting for 59.7%, 55.8%, and 67.3%, respectively. Additionally, a substantial reduction in PM within the 0.1–2.5 μm fraction was observed, with values of 40.3%, 44.2%, and 32.7% for *P. orientalis*, *M. alba*, and *U. minor*, respectively. Notably, *A. altissima* exhibited a 100% average presence of PM in the 0.1–2.5 μm fraction in Site B, surpassing other species. Moreover, *R. pseudoacacia* demonstrated 65% presence in the 0.1–2.5 μm fraction and 35% in PM exceeding 2.5 μm (Table 1).

### 3.3. Variability in Heavy Metal Accumulation Capacity among Diverse Tree Species in Two Distinct Sites

This study shows the contrasting abilities of different tree species to amass heavy metals at Sites A and B. The concentrations of nickel (Ni), cadmium (Cd), lead (Pb), and iron (Fe) (mg/kg dry weight) within tree samples from each site are graphically depicted in Figure 4.

In Site A, *A. altissima* demonstrated significantly higher nickel (Ni) accumulation, with a mean value of $30.91 \pm 4.1$, while at Site B, *R. pseudoacacia* exhibited superior Ni deposition, averaging $38.88 \pm 1.6$. Conversely, both Sites A and B revealed lower Ni accumulation in *P. orientalis*, with mean values of $23.94 \pm 4.5$ and $22.89 \pm 1.7$, respectively. At Site A, *U. minor*, *M. alba*, and *R. pseudoacacia* displayed intermediate Ni accumulation, while at Site B, following *R. pseudoacacia*, *U. minor*, *M. alba*, and *A. altissima* exhibited higher Ni uptake (Figure 4).

In terms of cadmium (Cd) and lead (Pb) deposition, no significant variations were observed among the species at Site A. In contrast, at Site B, *R. pseudoacacia* exhibited significantly higher Cd deposition with a concentration of $0.83 \pm 0.1$, as well as elevated Pb deposition at $190.5 \pm 12.4$ (Figure 4).

At Site B, *A. altissima* recorded the lowest Cd accumulation, measuring $0.15 \pm 0.1$, while *P. orientalis*, *U. minor*, and *M. alba* displayed intermediate Cd uptake. Furthermore, in Site B, *P. orientalis* exhibited the lowest Pb deposition ($114.51 \pm 11.1$), followed by *U. minor* and *M. alba*, which showed higher Pb accumulation after *R. pseudoacacia*. *A. altissima* displayed intermediate Pb uptake at Site B (Figure 4).

Furthermore, *A. altissima* displayed significantly higher iron (Fe) deposition in both study locations, with Site A recording a mean of $29,229.8 \pm 1242.6$ and Site B recording a mean of $29,144.8 \pm 5443.8$. In Site B, *R. pseudoacacia* ($33,529.1 \pm 2067.2$) and *M. alba* ($28,588.9 \pm 1915.6$) exhibited elevated Fe deposition, surpassing *U. minor* and *P. orientalis*. In Site A, following *A. altissima*, *P. orientalis* displayed higher Fe uptake, with a mean value of $24,150.2 \pm 4082.5$, while *R. pseudoacacia*, *U. minor*, and *M. alba* demonstrated intermediate levels of Fe accumulation (Figure 4).

### 3.4. Evaluation of EC, OC, and TC Deposition Capacities of Various Tree Species at Two Distinct Sites

The investigation into the deposition of Elemental Carbon (EC), Organic Carbon (OC), and Total Carbon (TC) on the leaf surfaces of different tree species at Sites A and B unveiled significant variations among *R. pseudoacacia*, *M. alba*, *A. altissima*, *U. minor*, and *P. orientalis* (Table 2).

At Site A, the deposition sequence of EC was as follows: *U. minor* > *P. orientalis* > *R. pseudoacacia* = *A. altissima* > *M. alba*. Notably, *U. minor* demonstrated significantly higher EC deposition, while *M. alba* exhibited the lowest. In Site B, *P. orientalis* and *R. pseudoacacia* exhibited the highest EC deposition, while *M. alba* had the lowest, and *U. minor* and *A. altissima* demonstrated intermediate levels (Table 2).

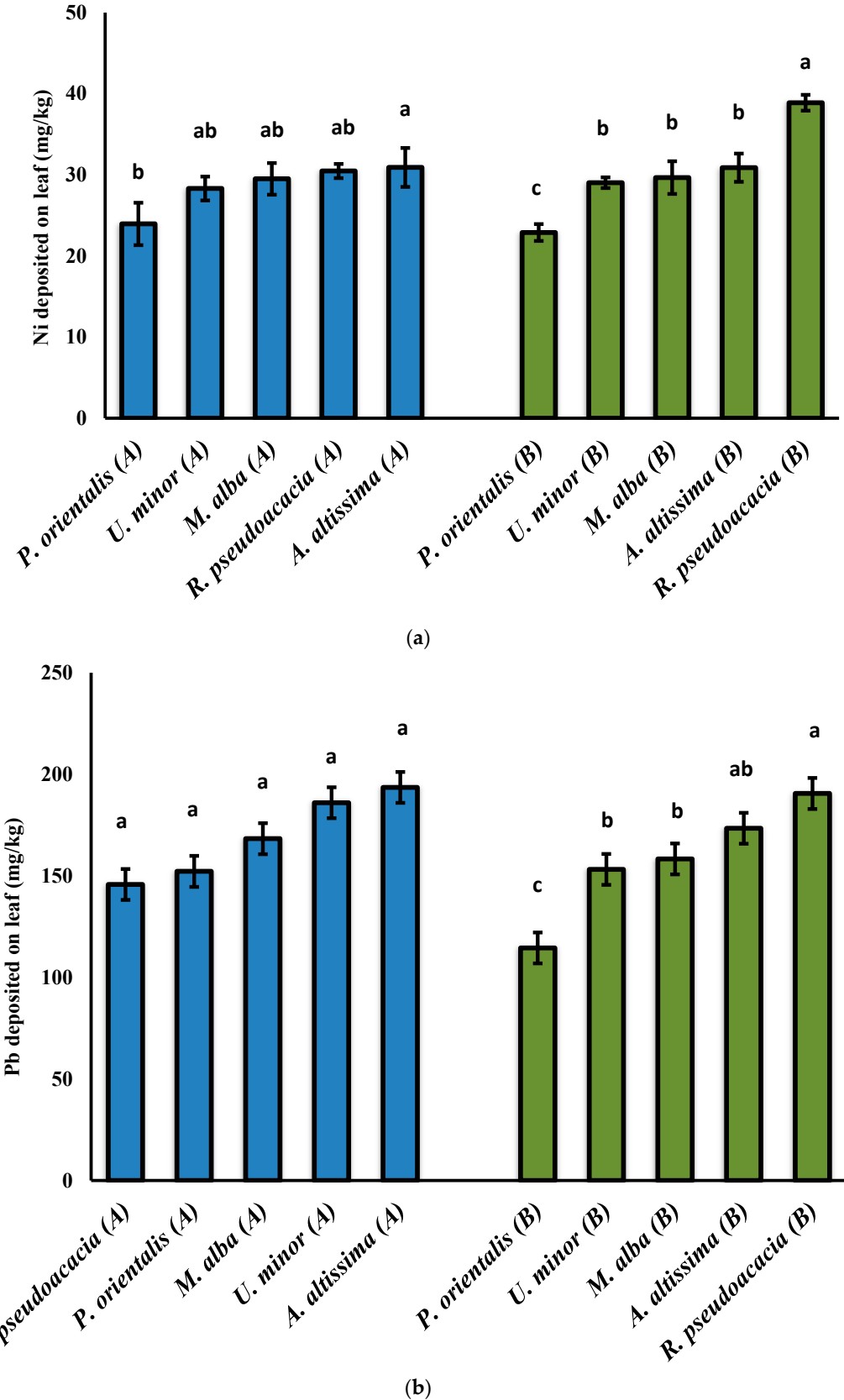

**Figure 4.** *Cont.*

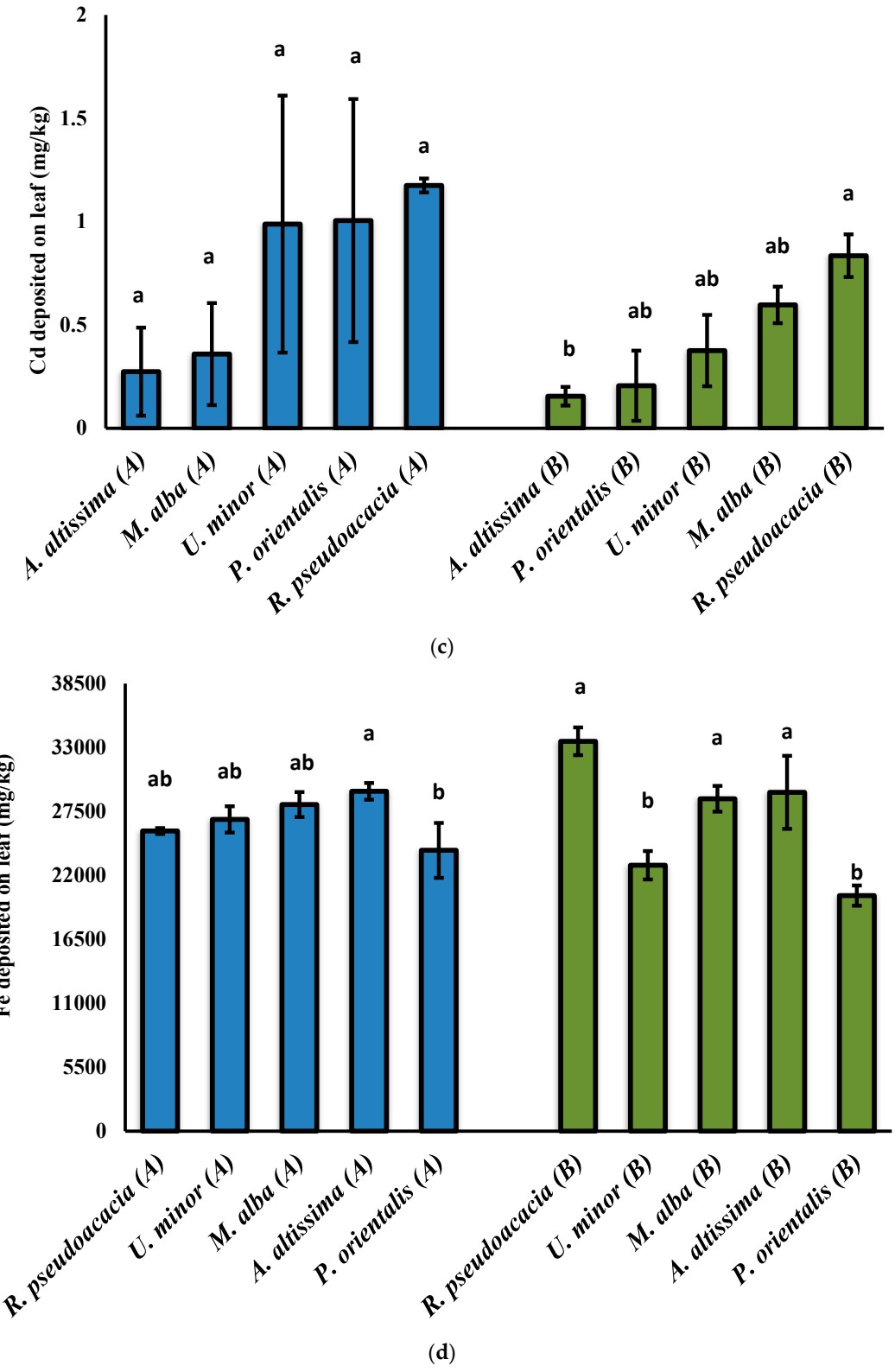

**Figure 4.** Deposition of Ni (**a**), Pb (**b**), Cd (**c**), and Fe (**d**) on the leaves of five tree species in Sites A and B. Data represented as mean ± standard deviation. Significantly different bars are marked with letters ($p < 0.05$). Different lowercase letters indicate significant differences.

**Table 2.** Mean deposition of TC, OC, and EC on the leaves of tree species at sites A and B.

| Species | Carbon Deposition in Site A (mg/cm$^2$) | | | Carbon Deposition in Site B (mg/cm$^2$) | | |
|---|---|---|---|---|---|---|
| | TC | OC | EC | TC | OC | EC |
| *R. pseudoacacia* | 1.204 ± 0.127 c | 0.400 ± 0.083 b | 0.803 ± 0.049 a | 0.966 ± 0.354 b | 0.334 ± 0.111 a | 0.632 ± 0.246 ab |
| *U. minor* | 2.570 ± 0.576 c | 0.925 ± 0.173 b | 1.645 ± 0.414 a | 0.845 ± 0.224 b | 0.332 ± 0.065 a | 0.513 ± 0.147 a |
| *M. alba* | 0.484 ± 0.170 b | 0.197 ± 0.060 a | 0.286 ± 0.116 a | 0.655 ± 0.087 b | 0.290 ± 0.047 a | 0.365 ± 0.071 a |
| *A. altissima* | 1.115 ± 0.113 c | 0.392 ± 0.062 b | 0.722 ± 0.060 a | 0.846 ± 0.202 b | 0.309 ± 0.102 ab | 0.524 ± 0.226 a |
| *P. orientalis* | 2.057 ± 0.467 c | 0.780 ± 0.284 b | 1.276 ± 0.319 a | 1.092 ± 0.311 c | 0.358 ± 0.050 b | 0.733 ± 0.178 a |

Data presented as mean values (mg/cm$^2$) ± standard deviation. Different lowercase letters in the same row indicate significant differences ($p < 0.05$).

Regarding OC deposition at Site A, *U. minor* and *P. orientalis* exhibited significantly higher levels, while *M. alba* displayed the lowest, and *A. altissima* and *R. pseudoacacia* fell in between. In Site B, no significant differences were observed among species regarding OC deposition (Table 2).

In the case of TC deposition at Site A, the order was found to be: *U. minor* > *P. orientalis* > *A. altissima* = *R. pseudoacacia* > *M. alba*. Similarly, at Site B, *P. orientalis* demonstrated significantly higher TC deposition, *M. alba* exhibited significantly lower TC deposition, and *A. altissima*, *U. minor*, and *R. pseudoacacia* displayed intermediate values (Table 2).

Table 2 highlights that Site A and Site B showed a significantly higher amount of EC deposition compared to OC deposition. The table indicates that EC was the major source of pollution in Site A, while both OC and EC contributed significantly to pollution in Site B.

## 4. Discussion

### 4.1. Effect of Species and Regions on PM Deposition

Tree species exhibit variations in their capacity for PM deposition. This variability is influenced by several factors, including leaf morphology, canopy structure and size, ambient air particle concentrations, particle size distribution, traffic volume, and seasonal variations [29,43,47,53–58].

Leaves with larger surface areas and longer petioles have been found to deposit more PM, whereas leaves with intricate shapes, such as lobed leaves, and rough surfaces tend to capture a greater amount of PM per square centimeter [59]. Moreover, taller trees with larger leaves exhibit enhanced efficiency in absorbing small particles through processes like diffusion, facilitated by turbulent airflow and impaction. Additionally, leaves adorned with trichomes (leaf hairiness) have proven to be significant in capturing and retaining PM [60–62]. The deposition of PM is intricately linked to seasons, particularly during spring and summer, when crowded streets experience elevated PM concentrations owing to frequent vehicular stops and accelerations [48,56,63].

In agreement with Sharma et al. (2020), our study also found a higher deposition of PM in traffic-impacted areas, specifically for *M. alba* [64]. Similarly, Gupta et al. (2016) conducted research in the National Capital Territory of Delhi and reported that *M. alba* exhibited greater PM fall fluxes compared to *Terminalia arjuna*, attributable to the greater roughness of *M. alba*'s foliar surface [65]. Chaudhary et al. (2018) investigated the dust removal efficiency of roadside trees and identified air pollution-tolerant tree species. Their findings reinforce the idea that leaves with larger surface areas and longer petioles are more adept at depositing dust [58], whereas leaves with compound phyllotaxy, alternate arrangements, and rough surfaces exhibit heightened dust-capturing capabilities [58,64–66]. Tallis et al. (2011) posited that larger leaves are more effective at absorbing small particles via diffusion, thanks to turbulent air flow and impaction [67]. However, Weerakkody et al. (2018) observed that smaller leaves with intricate shapes, such as lobed leaves, are more efficient in capturing and retaining PM, with leaf hairiness (the presence of trichomes) emerging as the pivotal factor [68].

Notably, our findings align with studies conducted in diverse geographical regions. In northern China, *A. altissima* and *M. alba* displayed significantly lower total surface PM

accumulation and in-wax PM accumulation, respectively [69]. Similarly, studies in Poland supported the finding that *P. orientalis* exhibited the highest PM accumulation capacity on its leaves. These consistent results across distinct regions underscore the robustness of our present research [48]. Koczak et al. (2021) likewise reported analogous results, indicating that *P. orientalis* stands out as the most efficient phytoremediation species [48]. The uniformity of these findings across diverse regions amplifies the significance of our research.

Moreover, the disparities in the capacity of tree species to capture particulate matter can be attributed to fluctuations in weather conditions associated with diverse traffic patterns and the higher wind speeds characteristic of densely trafficked urban areas. Elevated wind speeds bolster the deposition velocity of PM on roadside vegetation, leading to heightened levels of PM accumulation [70–74]. Chowdhury et al. (2022) conducted a study in Bangladesh to investigate particulate matter levels across six different sites. They discovered that roadside locations with less greenspace exhibited the highest atmospheric PM concentration, whereas areas with more greenspace, such as residential and park areas, displayed the lowest PM concentration. However, despite disparities in greenspace, there were no significant variations in atmospheric PM concentration levels among the sites [75].

In the present study, we observed that on both sites A and B, none of the species exceeded the fraction of particulate matter less than 0.1 μm (PM < 0.1), and $PM_{2.5}$ constituted the most significant percentage of captured particulate matter. These findings corroborate previous research by Dang et al. (2022) in the southwest of Hangzhou, where plant leaves primarily trapped coarse particles (PM > 10 μm), following the order $PM_{>10} > PM_{10} > PM_{2.5}$, where large particles constituted the majority of PM retained by plant leaves in a roadside environment, reinforcing our observations [76]. Elevated concentrations of $PM_{2.5}$ and $PM_{10}$ typically result from anthropogenic activities, including the generation of secondary particles through the combustion of hydrocarbon fuels. These higher values may also be linked to vehicle emissions and the formation of secondary particles through photochemical processes occurring at elevated temperatures [77]. In a study conducted by Biswas et al. (2020), the research delves into the consequences of rapid economic growth and urbanization on air quality in Kolkata and Siliguri, concentrating on three specific zones. The findings indicate significant concentrations of air pollutants, notably $PM_{2.5}$ and $PM_{10}$, with peak levels observed during the post-monsoon season in the Rabindra Bharati University zone in Kolkata. Monthly analyses of pollutant variations across the study areas underscore the impact of meteorological conditions and rising temperatures stemming from a variety of pollution sources [10]. In Zapletal et al.'s (2022) study, $PM_{10}$ aerosols were analyzed for PAHs, hopanes, and elements during the monsoon and pre-monsoon. Pre-monsoon $PM_{10}$ concentrations exceeded health limits, with major local sources identified as traffic, biomass burning, and coal combustion, plus contributions from waste incineration and a nearby cement factory. Diagnostic ratios indicated biomass and coal combustion impact post-monsoon pollution, while pre-monsoon saw petroleum combustion dominance. This study suggested pre-monsoon $PM_{10}$ originated locally or regionally, contrasting with monsoon transport from larger distances [11]. The intricate chemical composition of PM, encompassing carbonaceous species, inorganic ions, elements, hopanes, and PAHs, varies depending on location and emission sources. In the context of developing countries, there is a crucial need to supplement the classification of sources of carbon, PAHs, hopanes, and elements, emphasizing their impact on human health. Considering these variations in PM deposition among tree species in different particle size fractions, this becomes highly significant for air quality management. Identifying tree species with superior capabilities to capture specific PM size fractions holds immense value for informing urban greening and pollution mitigation strategies.

In discussing the outcomes of this study, it becomes evident that various tree species exhibit diverse PM deposition capacities within urban environments. Notably, *U. minor* and *M. alba* emerge as promising candidates for phytoremediation efforts in polluted areas, demonstrating substantial potential for capturing particulate matter on their leaves. These

findings emphasize the critical importance of thoughtful tree species selection in urban planning to effectively mitigate air pollution and enhance overall air quality.

However, it is crucial to acknowledge this study's limitations. The methodological choices used to measure PM deposition and the sample size of tree species might influence the results. Additionally, the focus on two specific sites could constrain the generalizability of our findings. Further investigations are warranted to explore specific causal relationships and seasonal variations in PM deposition, which are essential for reinforcing the credibility of these results.

Looking ahead, future research endeavors could delve into assessing the long-term efficacy of different tree species in capturing PM and their overall contributions to enhancing air quality in urban environments. Exploring the combined effects of various tree species and their interactions with local meteorological conditions could provide valuable insights for optimizing urban greening strategies aimed at ameliorating air pollution.

### 4.2. Effect of Species and Regions on Heavy Metal Accumulation

Plants, through their leaves, can accumulate heavy metals, with particulate matter serving as the primary carrier of these contaminants. The accumulation of heavy metals in plant leaves is influenced by several factors, including the plant species, leaf structure, plant growth status, leaf structural properties, the heavy metal content within PM, and traffic volume [61,63,73,76]. Roy et al. (2020) conducted tests within the same environment, revealing varying degrees of heavy metal accumulation due to differences in the ability of five plant species to accumulate heavy metals [78].

Jia et al. (2021) observed a significant positive correlation between the concentration of toxic metals in leaves and the retention of dust. Leaves with a relatively rough surface, such as those of *L. chinense*, displayed enhanced particle absorption and, consequently, higher levels of heavy metal accumulation [1]. The research by Hu et al. (2017), which investigated the bioaccumulation of heavy metals in plant leaves across various regions in China, like our results, their study indicated higher heavy metal levels in areas with heavy traffic, implying a relationship between pollution levels and traffic volume [79].

Of the tree species examined, *R. pseudoacacia* emerged as the most effective in reducing heavy metal deposition on leaf surfaces. This finding aligns with a study by Isinkaralar et al. (2022), which also identified *R. pseudoacacia* as a superior choice for assessing atmospheric heavy metal levels [80]. The unique leaf characteristics of *R. pseudoacacia*, characterized by fewer apertures and cracks, likely contribute to its efficient surface adsorption of heavy metals, potentially inhibiting their internal absorption within the leaf [1].

Our results further coincide with a study conducted by Alahabadi et al. (2017), which focused on the heavy metal accumulation capacity of various tree species. Like our study, they found that *M. alba* exhibited the highest bioaccumulation capacity for heavy metals, underscoring its significant potential for capturing heavy metals [81].

Furthermore, studies conducted in Tehran, Iran, demonstrated that *R. pseudoacacia* exhibited the highest accumulation of Cd and Pb among the leaves and shoots of *P. orientalis*, *R. pseudoacacia*, and *Fraxinus rotundifolia* [82]. Another study in Isfahan, Iran, indicated that *P. orientalis* could serve as an indicator of heavy metal pollution, except for Pb, which exhibited lower concentrations in leaves due to its nonessential nature, unlike Ni and Fe, which are essential for plant growth and enzyme activities affecting protein synthesis [23].

It is crucial to acknowledge the complexity of heavy metal contamination in urban areas, particularly from traffic emissions. Cd is released from engine and brake pad wear, while Pb is emitted from exhaust emissions. PM particles containing Fe are prevalent due to road activity and abrasion caused by heavy vehicles [48,79,83]. The higher deposition of heavy metals in site B can be attributed to the greater traffic volume and car exhaust emissions. Site B, classified as a traffic road (highway), exhibited higher heavy metal deposition compared to Site A, which is a residential and commercial area with relatively lower heavy metal accumulation.

In our prior research, we assessed heavy metal concentrations and pollution levels in PM deposition on the leaf using enrichment factors (EF), geoaccumulation index (Igeo), modified degrees of contamination (mCd), and a new weighted degree of contamination factor (wCd). EF values indicated significant enrichment, attributing PM pollution to anthropogenic activities. Igeo values pinpointed Cd (80%–97%) and Pb (100%) as major pollutants in DS. While mCd results suggested over 67% of samples were unpolluted, a notable discrepancy emerged, with Igeo indicating high pollution for Pb and Cd [46].

These findings hold significant implications for urban planning strategies aimed at addressing heavy metal pollution. However, it is essential to recognize that contrasting results from other studies underscore the need for further research to comprehend site-specific factors influencing heavy metal accumulation in urban environments. Given the diverse sources and levels of heavy metal contamination in urban areas, interdisciplinary investigations are crucial for developing effective pollution control strategies and promoting sustainable urban development.

### 4.3. Effect of Species and Regions on Carbon Accumulation

In this study, we explored the deposition of EC, OC, and TC on the leaf surfaces of various tree species at sites A and B. The findings provide valuable insights into the interactions between these tree species and air quality, shedding light on their abilities to capture particulate matter. To place our results into context, we also contrasted our findings with previous research, offering a broader understanding of the intricate relationship between carbonaceous pollution and tree species.

In comparison to previous research, our findings offer a distinct perspective. Feng et al. (2009) observed higher levels of OC compared to EC in both rural and urban areas. This distinction is particularly pronounced in rural regions where carbonaceous pollution, stemming from vehicular exhaust, coal smoke, and kitchen emissions, takes on a more prominent role [84]. Similarly, Ye et al. (2003) identified significantly elevated OC levels in two sites in China, surpassing EC levels. These findings were attributed to emissions from diesel vehicles and older gasoline-powered vehicles [85]. In South Korea, Kim et al. (2022) categorized EC and OC as the third sources of pollution for mobile and coal-fired sources, respectively, in urban areas [86]. Jones et al. (2005) reported a different pattern, with higher levels of EC compared to OC at kerbside sites and a notable increase in EC levels during the week compared to weekends in North Kensington sites [87]. Observations in southeast Osaka, Japan, along three roadsides showed elevated Elemental Carbon levels in comparison to Organic Carbon levels, potentially due to diesel vehicle emissions [88].

In this specific study, we observed the unique abilities of *U. minor* and *M. alba* to effectively capture particulate matter on their leaves within the urban sites under investigation. The variation in deposition capacities among tree species was influenced by factors such as PM concentration levels, weather conditions, and wind speeds. A significant finding from our study was the absence of substantial differences between the two sites concerning OC, EC, and TC deposition, indicating consistent source emission rates.

These results underscore the crucial role that careful selection of tree species plays in urban planning to combat air pollution and enhance air quality. However, there is a need for additional research to delve into the underlying causal relationships and address potential limitations in our understanding of these complex interactions.

The unique aspect of this study is its comprehensive assessment of five tree species in Tehran, focusing on their capacity to capture particulate matter, heavy metals, and various forms of carbon, thereby offering invaluable insights into their potential as natural air purification agents. It is worth highlighting that, as of now, no prior research has undertaken a comparative analysis to evaluate the effectiveness of different tree species in capturing and retaining particulate matter, heavy metals, elemental carbon, organic carbon, and total carbon in the context of Iran.

The outcomes of this study have significant implications for urban planning and green space design, not only in Tehran but also in other major cities facing similar air

pollution challenges. Identifying tree species with higher PM deposition capacities can help in the selection of capable species for urban afforestation, improving air quality in heavily polluted areas. Moreover, awareness of the capacity of specific tree species to capture heavy metals and carbon creates strategies for mitigating the adverse effects of heavy metals and carbon pollution in urban environments.

In a fast-growing urbanization process all over the world where air pollution poses an escalating threat to public health and the environment, the findings of this study contribute valuable information to the growing body of research on urban greening as a sustainable solution to combat air pollution. Integrating effective phytodepuration strategies into urban planning and environmental management can pave the way for building healthier, more resilient, and sustainable cities for present and future generations. Ultimately, fostering green urban environments can not only enhance air quality but also elevate the overall quality of life for urban inhabitants, creating livable, vibrant, and environmentally friendly cities.

## 5. Conclusions

In this study, we investigated the particulate matter (PM) deposition capacity of different tree species at two sites within Tehran, Iran. The findings shed light on the potential of various tree species to capture PM, which is a critical aspect of urban air quality management and pollution mitigation.

Our results revealed significant differences in PM deposition capacity among the studied tree species. *U. minor* and *M. alba* demonstrated strong abilities to capture particulate matter on their leaves, making them promising candidates for urban greening and air pollution reduction initiatives. On the other hand, *R. pseudoacacia* and *A. altissima* exhibited the lowest capacity for PM deposition at both sites.

Furthermore, we explored the deposition of heavy metals, such as Ni, Fe, Cd, and Pb, on the leaves of different species. *U. minor* and *P. orientalis* displayed the highest potential for retaining PM 0.1–2.5 fractions, while *M. alba* exhibited greater retention of PM > 2.5 in site A. In site B, *A. altissima* demonstrated the strongest capacity to adsorb PM 0.1–2.5, while *U. minor* showed greater retention of PM > 2.5. These findings provide valuable insights into the roles of specific tree species in capturing particulate matter of varying sizes, which can inform targeted planting strategies for improving urban air quality.

Our study highlights the importance of selecting suitable tree species in urban areas to combat air pollution effectively. Planting species with higher PM deposition capacities can contribute significantly to reducing air pollution levels and enhancing the overall environmental quality of urban spaces. As cities continue to face the challenges of air pollution, incorporating the knowledge gained from this research into urban planning and green space design can contribute to a healthier and more sustainable urban environment.

While this study contributes valuable information, it is essential to recognize its limitations. The chosen study sites represent specific urban areas, and further research across various locations and climates could enhance the generalizability of the findings. Additionally, long-term monitoring and evaluation of the selected tree species' performance would provide a more comprehensive understanding of their potential as effective air quality improvement tools.

Overall, our findings emphasize the significant role of urban vegetation in mitigating air pollution and provide valuable guidance for urban planners, policymakers, and environmentalists in promoting sustainable and healthier urban environments. By integrating appropriate tree species into urban landscapes, we can take a step closer to building more resilient and greener cities for the benefit of current and future generations.

**Author Contributions:** Conceptualization, A.S., S.E. and M.M.; methodology, S.E., A.S. and M.M.; software, S.E. and A.S.; validation, A.S. and F.S.; formal analysis, S.E. and A.S.; investigation, S.E., A.S., M.M. and F.S.; resources, S.E. and A.S.; data curation, A.S., M.M. and F.S.; writing—original draft preparation, S.E.; writing—review and editing, S.E., A.S. and F.S.; visualization, A.S. and M.M.; supervision, A.S. and M.M. All authors have read and agreed to the published version of the manuscript.

**Funding:** This research received no external funding.

**Data Availability Statement:** The data will be made available by the authors on request.

**Conflicts of Interest:** The authors declare no conflicts of interest.

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
