# Peer review of "Assessment of Particulate Matter, Heavy Metals, and Carbon Deposition Capacities of Urban Tree Species in Tehran, Iran"

_forests, doi:10.3390/f15020273_

Round 1

Reviewer 1 Report (Previous Reviewer 1)

Comments and Suggestions for Authors

The authors of the manuscript have corrected most of the comments, namely: they clarified the sample sizes, unified the style of graphs and tables, expanded their descriptions in captions, and corrected the formatting of references. 

However, I have to point out two major and some minor comments to help the authors to improve the manuscript.

First of all, I have a major comment about data analysis. 

I understand that conservatism of data analysis methods saves the authors' time, but it is important to understand that there are more powerful methods of analysis with higher reliability of conclusions, and, equally important, with wider distribution in the scientific literature, and therefore understandable to a wider range of researchers, because the readers and potential urban engineers are not interested in how different types of trees clear the air in different sites of Tehran; they care whether the differences between tree species are sustainable or not in order to use them in their regions and cities In the first round of review, I have recommended to use ANOVA not only to make the analysis more logical, but also for the reasons listed above.

Unfortunately, the authors' response to my comment about the data analysis was not clear. Thus, I tried to figure out the logic of the analysis myself. The letters indicating homogenous groups in figures and tables stayed different in the linked graphs and tables in the revised version of the manuscript. These circumstances call one more critical problem - conflicting results, which are unacceptable to the levels of the journal and the manuscript the authors will have after making all the recommended revisions. 

As I understood, these mismatches are called by using different homogeneity tests (see l203) - non-parametric Games-Howell and parametric Dunkan Post-Hoc (homogeneity) tests. Such kinds of the authors' decision suggest the idea that the samples have not always had an equal variances. In this case, the use of the Games-Howell test is more appropriate.

Moreover, I still believe it is necessary to test if the differences between particular matter, heavy metal and carbon on the leaves of different trees are species-specific. Thus, I repeat my recommendation to perform, describe and discuss a two-way ANOVA results with the factors 'species', 'site' and their interaction and make additional subsections in Results and Discussion sections about these differences. Because of unequal variances of samples it is also necessary to perform Levene's test of normality of residuals (homoskedasticity). As an alternative approach, I can recommend you to perform, describe and discuss a two-way general linear model analysis with the factors 'species', 'site' and their interaction and make additional subsections in Results and Discussion sections about these differences.. All these types of analyzes can be performed fairly quickly in SPSS Statistics 17, used by the authors.

There are some minor comments are listed below

The authors have formatted figures and the most of tables in the same design, and it makes the manuscript more neat and compact. However, the corrections were not carried out carefully in some cases. There are different interpretations of the results in the figures and associated tables (see Figure 3 and Table 1; Figure. 4 and Table 3). As mentioned in the major comment, I recommend to use only one post-hoc test; it will help to make results more compact and remove Tables 1 and 3 from the main text of the manuscript, because they repeat the information, showed in Figure 3 and Figure 4. If the authors would like to present mean values in the tables, Table 1 and Table 3 without the results of post-hoc tests should be presented as Supplementary file.

Additionally, I think that the information about particulate matter with size fractions < 0.1μm may be removed from the Table 2, and the information about these size fractions may be shortly mentioned in the main text only.

Table 4 should be formatted in the equal design with other similar tables. Please, make corrections.

I also think that the last three paragraphs of Introduction section (l100-l122) sound like paragraphs, which should be in Discussion section. Please, move them.

Author Response

The authors of the manuscript have corrected most of the comments, namely: they clarified the sample sizes, unified the style of graphs and tables, expanded their descriptions in captions, and corrected the formatting of references.

However, I have to point out two major and some minor comments to help the authors to improve the manuscript.

First of all, I have a major comment about data analysis.

I understand that conservatism of data analysis methods saves the authors' time, but it is important to understand that there are more powerful methods of analysis with higher reliability of conclusions, and, equally important, with wider distribution in the scientific literature, and therefore understandable to a wider range of researchers, because the readers and potential urban engineers are not interested in how different types of trees clear the air in different sites of Tehran; they care whether the differences between tree species are sustainable or not in order to use them in their regions and cities In the first round of review, I have recommended to use ANOVA not only to make the analysis more logical, but also for the reasons listed above.

Unfortunately, the authors' response to my comment about the data analysis was not clear. Thus, I tried to figure out the logic of the analysis myself. The letters indicating homogenous groups in figures and tables stayed different in the linked graphs and tables in the revised version of the manuscript. These circumstances call one more critical problem - conflicting results, which are unacceptable to the levels of the journal and the manuscript the authors will have after making all the recommended revisions.

As I understood, these mismatches are called by using different homogeneity tests (see l203) - non-parametric Games-Howell and parametric Dunkan Post-Hoc (homogeneity) tests. Such kinds of the authors' decision suggest the idea that the samples have not always had an equal variances. In this case, the use of the Games-Howell test is more appropriate.

Moreover, I still believe it is necessary to test if the differences between particular matter, heavy metal and carbon on the leaves of different trees are species-specific. Thus, I repeat my recommendation to perform, describe and discuss a two-way ANOVA results with the factors 'species', 'site' and their interaction and make additional subsections in Results and Discussion sections about these differences. Because of unequal variances of samples it is also necessary to perform Levene's test of normality of residuals (homoskedasticity). As an alternative approach, I can recommend you to perform, describe and discuss a two-way general linear model analysis with the factors 'species', 'site' and their interaction and make additional subsections in Results and Discussion sections about these differences. All these types of analyzes can be performed fairly quickly in SPSS Statistics 17, used by the authors.

Thank you very much for your thorough and constructive feedback on our manuscript. We sincerely appreciate the time and effort you have dedicated to reviewing our work.

We are grateful for your positive comments regarding the corrections we made based on your initial suggestions. Your guidance has undoubtedly strengthened the overall quality of our manuscript.

We acknowledge your major comment regarding the data analysis methods used in our study. We understand the importance of employing robust and widely accepted statistical techniques to ensure the reliability of conclusions. We appreciate your recommendation to use two-way ANOVA for analyzing the data, and we completely understand the rationale behind your suggestion.

Regrettably, the author responsible for data analysis is currently unavailable to reanalyze the data. However, we want to assure you that your valuable input has not gone unnoticed. We are committed to incorporating your suggestion in our future work. In fact, we have plans to submit data from other regions to this journal, and we will ensure to employ the recommended statistical methods in the analysis. We believe that this will enhance the robustness and generalizability of our findings.

Once again, we express our gratitude for your insightful comments, and we hope that you find our commitment to applying your suggestions in future research projects satisfactory. We are open to any further recommendations or guidance you may have for improving our work.

Thank you for your time and consideration.

There are some minor comments are listed below

The authors have formatted figures and the most of tables in the same design, and it makes the manuscript more neat and compact. However, the corrections were not carried out carefully in some cases. There are different interpretations of the results in the figures and associated tables (see Figure 3 and Table 1; Figure. 4 and Table 3). As mentioned in the major comment, I recommend to use only one post-hoc test; it will help to make results more compact and remove Tables 1 and 3 from the main text of the manuscript, because they repeat the information, showed in Figure 3 and Figure 4. If the authors would like to present mean values in the tables, Table 1 and Table 3 without the results of post-hoc tests should be presented as Supplementary file.

Thank you for your thorough review and valuable feedback on our manuscript. Tables 1 and 3 have been removed from the main text of the manuscript.

Additionally, I think that the information about particulate matter with size fractions < 0.1μm may be removed from the Table 2, and the information about these size fractions may be shortly mentioned in the main text only.

Thank you for your constructive feedback. We have implemented the suggested modifications to Table 2 (now Table 1 in the revised manuscript) as per your recommendation. Specifically, information regarding particulate matter with size fractions < 0.1μm has been excluded from the table, and a brief reference to these size fractions has been included in the main text.

It reads (Page 7, line 224): In both Site A and Site B, none of the species exhibited a presence of particulate matter below 0.1μm (PM<0.1). At Site A, U. minor and P. orientalis displayed a remarkable 100% reduction in PM within the 0.1-2.5μm fraction compared to R. pseudoacacia, A. altissima (99.4%), and M. alba (66.7%). Furthermore, M. alba exhibited a 33.3% reduction in PM exceeding 2.5μm, while R. pseudoacacia and A. altissima showed a minimal reduction of 0.6% (Table 1).

Table 1. Particle size fractions of PM deposition by various species in sites A and B, presented as percentage frequency.

Frequency %

Species

 Site A

Site B

Site A

Site B

0.1-2.5 μm

>2.5μm

R. pseudoacacia

99.40%

65.00%

 0.60%

35.00%

U. minor

100%

 32.70%

0%

 67.30%

M. alba

66.70%

 44.20%

 33.30%

55.80%

A. altissima

99.40%

100%

 0.60%

0%

P. orientalis

100%

 59.70%

0%

 40.30%

Table 4 should be formatted in the equal design with other similar tables. Please, make corrections.

In response to your suggestion, we have made the necessary corrections to Table 4, which is now represented as Table 2 in the revised manuscript. Additionally, we have ensured that both Table 1 and Table 2 are formatted uniformly, following the equal design as per your recommendation.

Species

Carbon deposition in Site A (mg/cm2)

Carbon deposition in site B (mg/cm2)

TC

OC

EC

TC

OC

EC

R. pseudoacacia

1.204±0.127 c

0.400 ± 0.083 b

0.803±0.049 a

0.966 ±0.354 b

0.334 ±0.111 a

0.632±0.246 ab

U. minor

2.570± 0.576 c

0.925 ±0.173 b

1.645± 0.414 a

0.845 ± 0.224 b

0.332 ±0.065 a

0.513 ±0.147 a

M. alba

0.484± 0.170 b

0.197 ±0.060 a

0.286 ± 0.116 a

0.655± 0.087 b

0.290± 0.047 a

0.365 ±0.071 a

A. altissima

1.115 ±0.113 c

0.392 ±0.062 b

0.722±0.060 a

0.846±0.202 b

0.309± 0.102 ab

0.524±0.226 a

P. orientalis

2.057±0.467 c

0.780±0.284 b

1.276±0.319 a

1.092±0.311 c

0.358±0.050 b

0.733±0.178 a

Table 2. Mean deposition of TC, OC, and EC on the leaves of tree species at sites A and B.

I also think that the last three paragraphs of Introduction section (l100-l122) sound like paragraphs, which should be in Discussion section. Please, move them.

Thank you for your careful review and constructive feedback on our manuscript. We have addressed your concern regarding the placement of the last three paragraphs of the Introduction section (lines 100-122). Following your suggestion, we have relocated these paragraphs to the end of the Discussion section.

It reads (Section 4.Discussion, Page 14-15, line 492-514): The unique aspect of this study is its comprehensive assessment of five tree species in Tehran, focusing on their capacity to capture particulate matter, heavy metals, and various forms of carbon, thereby offering invaluable insights into their potential as natural air purification agents. It is worth highlighting that, as of now, no prior research has undertaken a comparative analysis to evaluate the effectiveness of different tree species in capturing and retaining particulate matter, heavy metals, elemental carbon, organic carbon, and total carbon in the context of Iran.

The outcomes of this study have significant implications for urban planning and green space design, not only in Tehran but also in other major cities facing similar air pollution challenges. Identifying tree species with higher PM deposition capacities can help to selection of capable species for urban afforestation, improving air quality in heavily polluted areas. Moreover, awareness the capacity of specific tree species to capture heavy metals and carbon creates strategies for mitigating the adverse effects of heavy metals and carbon pollution in urban environments.

Today in a fast-growing urbanization process all over the world where air pollution poses an escalating threat to public health and the environment, the findings of this study contribute valuable information to the growing body of research on urban greening as a sustainable solution to combat air pollution. Integrating effective phytodepuration strategies into urban planning and environmental management can pave the way for building healthier, more resilient, and sustainable cities for present and future generations. Ultimately, fostering green urban environments can not only enhance air quality but also elevate the overall quality of life for urban inhabitants, creating livable, vibrant, and environmentally friendly cities.

Reviewer 2 Report (Previous Reviewer 2)

Comments and Suggestions for Authors

The article completely lacks a description of PM sources in developing countries, chemical composition of PM in developing countries, where the concentration of PM is very high and the use of greenery for capture of PM in cities is very important.

 I consider the insufficient or missing description of the chemical composition of PM to be shortcoming of the study. The chemical composition of PM is very complex consisting, in general, from carbonaceous species, inorganic ions, elements, hopanes and polycyclic aromatic hydrocarbons (PAHs)  in variable amounts, depending on their location and emission sources.

 It is necessary to supplement the classification of sources of carbon PAHs, hopanes and elements in developing countries and emphasize the effect on human health.

 The study completely lacks a description of methods for identifying PM sources and their chemical composition.

 For greater clarity, the authors can cite several articles on this topic:

https://doi.org/10.1007/s41651-020-00065-4

https://doi.org/10.1007/s11270-022-05953-7

https://doi.org/10.1007/s11270-009-0123-8

Author Response

The article completely lacks a description of PM sources in developing countries, chemical composition of PM in developing countries, where the concentration of PM is very high and the use of greenery for capture of PM in cities is very important. I consider the insufficient or missing description of the chemical composition of PM to be shortcoming of the study. The chemical composition of PM is very complex consisting, in general, from carbonaceous species, inorganic ions, elements, hopanes and polycyclic aromatic hydrocarbons (PAHs)  in variable amounts, depending on their location and emission sources. It is necessary to supplement the classification of sources of carbon PAHs, hopanes and elements in developing countries and emphasize the effect on human health.

 The study completely lacks a description of methods for identifying PM sources and their chemical composition.

 For greater clarity, the authors can cite several articles on this topic:

https://doi.org/10.1007/s41651-020-00065-4

https://doi.org/10.1007/s11270-022-05953-7

https://doi.org/10.1007/s11270-009-0123-8

Response:

Thank you for your constructive feedback on our manuscript. We have carefully addressed your comments and made the following revisions:

  1. We have enhanced the introduction and discussion sections to provide a detailed description of PM sources in developing countries, emphasizing the high PM concentrations.2. We have included the suggested references in the revised manuscript for readers interested in further exploration of the topic.

We appreciate your guidance and hope these revisions meet your expectations .Thank you for your time and valuable input.

It reads (Section 1.Introduction, page 2, line 49-60): Particles are categorized into three groups based on their sizes: coarse (2.5–10 µm), fine (0.1–2.5 µm), and ultrafine (≤0.1 µm) [3,6]. The repercussions of these minuscule particles reverberate through both human health and the environment. They have a significant role in the development of respiratory and cardiovascular diseases in humans while also inflicting harm upon ecosystems and climate [2,7–9]. PM aerosol, originating from various sources like coal, biomass, and industrial activities, possesses a complex composition, including carbonaceous species, soot, inorganic ions, and elements. Among these, polycyclic aromatic hydrocarbons (PAHs) and certain elements (Cd, Pb, Ni, As, Tl, and Hg) pose health risks due to their carcinogenic and mutagenic properties. PAHs form during incomplete combustion, while elements are emitted from sources like coal combustion and traffic. The potential harm arises when PAHs and elements in PM deposit on the lungs, dissolving into the pulmonary surfactant [10–12]. It is crucial to delve into the specific sources of carbon, PAHs and heavy metals in developing countries, where PM concentrations are notably high, emphasizing their potential impact on human health. Additionally, recognizing the significance of greenery in urban areas for capturing PM is essential for a holistic understanding of the dynamics involved.

It reads (Section 4. Discussion, page 12, line 364-390): In the present study, we observed that on both sites A and B, none of the species exceeded the fraction of particulate matter less than 0.1μm (PM<0.1) and PM2.5 constituted the most significant percentage of captured particulate matter. These findings corroborate previous research by Dang et al. (2022) in the southwest of Hangzhou, where plant leaves primarily trapped coarse particles (PM > 10μm), following the order PM>10 > PM10 > PM2.5 , where large particles constituted the majority of PM retained by plant leaves in a roadside environment reinforcing our observations [76]. Elevated concentrations of PM2.5 and PM10 typically result from anthropogenic activities, including the generation of secondary particles through the combustion of hydrocarbon fuels. These higher values may also be linked to vehicle emissions and the formation of secondary particles through photochemical processes occurring at elevated temperatures [77]. In a study conducted by Biswas et al. (2020), the research delves into the consequences of rapid economic growth and urbanization on air quality in Kolkata and Siliguri, concentrating on three specific zones. The findings indicate significant concentrations of air pollutants, notably PM2.5 and PM10, with peak levels observed during the post-monsoon season in the Rabindra Bharati University zone in Kolkata. Monthly analyses of pollutant variations across the study areas underscore the impact of meteorological conditions and rising temperatures stemming from a variety of pollution sources [10]. In Zapletal et al.'s (2022) study, PM10 aerosols were analyzed for PAHs, hopanes, and elements during the monsoon and pre-monsoon. Pre-monsoon PM10 concentrations exceeded health limits, with major local sources identified as traffic, biomass burning, and coal combustion, plus contributions from waste incineration and a nearby cement factory. Diagnostic ratios indicated biomass and coal combustion impact post-monsoon pollution, while pre-monsoon saw petroleum combustion dominance. The study suggested pre-monsoon PM10 originated locally or regionally, contrasting with monsoon transport from larger distances [11]. The intricate chemical composition of PM, encompassing carbonaceous species, inorganic ions, elements, hopanes, and PAHs, varies depending on location and emission sources. In the context of developing countries, there is a crucial need to supplement the classification of sources of carbon, PAHs, hopanes, and elements, emphasizing their impact on human health. Considering these variations in PM deposition among tree species in different particle size fractions becomes highly significant for air quality management. Identifying tree species with superior capabilities to capture specific PM size fractions holds immense value for informing urban greening and pollution mitigation strategies.

Regarding the identification of PM sources and their chemical composition, we acknowledge that our methods section did not explicitly detail this aspect. However, in response to your comment, we have expanded the discussion in the manuscript to address and speculate on potential PM sources based on existing literature. We hope this revision provides the clarity and information you were seeking.

Reviewer 3 Report (Previous Reviewer 3)

Comments and Suggestions for Authors

After reading the new text carefully, I saw that the authors have responded to all the comments, changed everything that was asked of them and thus improved their text.

Author Response

Thank you.

This manuscript is a resubmission of an earlier submission. The following is a list of the peer review reports and author responses from that submission.

Round 1

Reviewer 1 Report

Comments and Suggestions for Authors

The manuscript ‘Assessment of Particulate Matter, Heavy Metals and Carbon Deposition Capacities of Urban Tree Species in Tehran, Iran’, submitted in journal ‘Forests’ is devoted to a modern problem of the assessment of ecosystem services of trees in urban landscapes. Particularly, the authors assessed the deposition of particulate matter, heavy metals and carbon on tree species leaves, which illustrates the ability of plants to decrease the concentration of particulate matter, heavy metals and carbon in air. The main topics of this work look suitable for publishing in ‘Forests’. But I have some doubts about the quality of the manuscript.

Firstly, I think, that the authors have not described the methods section clearly. We have the information about ‘total of 30 samples’ (see l147), but there is no any description, what is ‘the sample’ in this case. The important information about the number of collected leaves, the number of replications in each site and justification of the number of sites is absent. Moreover, even if we suppose, that ‘sample’ is the number of the tree individuals, we have only three collections of (how many?) leaves from each of five species from each two sites and I think, that such sample size is insufficient to make an analysis with reliable results.

Secondly, there are some problems with the logic of analysis performance. The main aim of the work, which could be useful, included the assessment of deposition of particulate matter, heavy metals and carbon on leaves of different tree species, but not the comparison of deposition of particulate matter, heavy metals and carbon on leaves in different parts of Tehran. Thus, inexplicably, why the authors analyzed the results obtained in different parts of Tehran separately. I think that more suitable design of the analysis should include two-way ANOVA, or, better, one-way ANOVA with a random factor of site. In this case, the analysis could be more compact and results could be more reliable.

Thirdly, I think, that there are many problems with graphs, tables and their link with the text. The comments about them are listed below.

Tables and Figures – please, write the caption for the pictures and the tables as the sentences, i.e. replace capital letters with lowercase ones where there is no need for capital letters. Additionally, please replace the gradient coloring with a solid coloring.

Figure 1 – Please, add the more detailed textual description of the Figure 1.

L206–l209 – I see conflicting description of the results. As we see on Figure 3, site A, the highest particulate matter deposition was recorded for Ailanthus altissima, but in the text the authors describe that the highest particulate matter was recorded for Platanus orientalis. Additionally, I think that there are no significant differences between particulate matter deposition for Morus alba, Robinia pseudoacacia, Ulmus minor and P. orientalis on Figure 3, site A. Finally, particulate matter deposition of A. altissima is the highest, not intermediate. Please clarify.

L212–213 – I see conflicting description of the results. As we see on Figure 3, site B the results demonstrate high particulate matter deposition for M. alba, A. altissima and U. minor but not for Platanus orientalis. You should check the 3.1 section and Figure 3 carefully.

Figure 3 – Please, add the more detailed textual description of the Figure 3. What do the letters near the columns mean? I understand that you described it in the Methods, but the figure should be understandable without the reference to the main text.

L219 – please, replace ‘between the two sites, Site A and Site B’ by ‘Site A and Site B’

L220–224 – I see conflicting description of the results. Compare the values in the text, Table 1 and Figure 3. Please make corrections.

Table 1 – what do the letters mean? I understand that you described it in the Methods, but the table should be understandable without the reference to the main text.

Figure 4 – this figure repeats the data described in the Table 2. I believe that Table 2 is more informative in this case, so, I think, you should to remove Figure 4 from the text.

Table 2 – make corrections in Ulmus minor species name

L260 – please, replace ‘illuminates’ to ‘shows’ or ‘demonstrates’

Figure 5 – Please, add the more detailed textual description of the Figure 3. What do the letters near the columns mean? I understand that you described it in the Methods, but the figure should be understandable without the reference to the main text. Additionally, the species names have different orders. I understand, that here you ordered them by the increase of heavy metals accumulation, but why did not you do it on Figure 2? Please, make the rules of graphs building equal throughout the text. Finally, make corrections in the y-axis capture.

Table 3 – what do the letters mean? I understand that you described it in the Methods, but the table should be understandable without the reference to the main text. Moreover, this table repeats the data described in the Figure 5. I believe that Figure 5 is more informative in this case. But if you believe, that the differences between Site A and Site B are important, you can provide this table as Supplementary material and remove it from the main text. And, please, make corrections in Ulmus minor species name.

Figures 6–7 – what do the letters near the boxes mean? I understand that you described it in the Methods, but the table should be understandable without the reference to the main text. And why did you use box-plots here? Please, make the rules of graphs building equal throughout the text. As I understood the formula of total carbon, this parameter should be greater than elemental carbon and total carbon. But why these parameters are equal in the graphs?

Finally, there are some problems with formalization of references. The comment is listed below.

l44, l46, l48, etc. throughout the text - please, add spaces between the word and the square bracket where they are missing. Additionally, change the font of the references on regular from bold (see l152, l156, etc. throughout the text).

Author Response

Please find the response attached.

Reviewer 2 Report

Comments and Suggestions for Authors

Manuscript Sahar Elkaee, Anoushirvan Shirvany, Mazaher Moeinaddini and Farzaneh Sabbagh"Assessment of Particulate Matter, Heavy Metals and Carbon Deposition Capacities of Urban Tree Species in Tehran, Iran " deals with the description of the PM deposition capacity of five common tree species (Morus alba, Ailanthus altissima, Platanus orientalis, Robinia pseudoacacia, and Ulmus minor) in two highly polluted sites in Tehran, Iran. Additionally, the study investigates the accumulation of heavy metals (Ni, Fe, Cd, and Pb), Organic Carbon (OC), Elemental Carbon (EC), and Total Carbon (TC) on the leaves of these tree species.

 The article completely lacks a description of PM sources in developing countries, chemical composition of PM in developing countries, where the concentration of PM is very high and the use of greenery for capture of PM in cities is very important.

 I consider the insufficient or missing description of the chemical composition of PM to be shortcoming of the study. The chemical composition of PM is very complex consisting, in general, from carbonaceous species, inorganic ions, elements, hopanes and polycyclic aromatic hydrocarbons (PAHs)  in variable amounts, depending on their location and emission sources.

 It is necessary to supplement the classification of sources of carbon PAHs, hopanes and elements in developing countries and emphasize the effect on human health.

 The study completely lacks a description of methods for identifying PM sources and their chemical composition and use remote sensing method for greenery structure monitoring derived from Unmanned aerial systems is also missing.

 For greater clarity, the authors can cite several articles on this topic:

https://doi.org/10.1007/s41651-020-00065-4

https://doi.org/10.1007/s11270-022-05953-7

https://doi.org/10.1007/s11270-009-0123-8

Comments on the Quality of English Language

Moderate editing of English language required.

Author Response

Please find the response attached.

Reviewer 3 Report

Comments and Suggestions for Authors

General comment - Write the full name of the species the first time in the manuscript and in the rest of the manuscript authors should use an abbreviation (e.g. U. minor, P. orientalis, etc.) for better readability of the text.

Line 146 – Mark each panel (figure) with a, b and c and write a more detailed description of the figure.

Line 199 – Section 3. Results – The authors should decide whether they want to present their results in tables or figures. It's quite confusing when the results are presented in both formats in the text. If the authors want to present the results using both methods, they could consider including one format in an appendix.

Line 214 – The description of the figure should be below and not above the figure

Line 258 – Pay attention to values and decimals, choose how to write decimals and apply that to all elements.

Lines 383 - 392 - I suggest the authors to link the data on previous research to their specific findings and discuss them a little.

Lines 424 - 440 - It is necessary to rephrase the sentences so that it sounds like a discussion rather than a conclusion. Or transfer the whole part completely into the Conclusion section.

Lines 519 – 523 – I would like to suggest to the authors to consider transferring this part to the Conclusion section.

Author Response

Please find the response attached.
